# OpenReview forum: "Curriculum-RLAIF: Curriculum Alignment with Reinforcement Learning from AI Feedback"
_ICLR.cc/2026/Conference — ICLR 2026 Conference Withdrawn Submission_

### Official Review · Reviewer_CQqb · 2025-10-29

**Soundness:** 3
**Presentation:** 2
**Contribution:** 3
**Rating:** 4
**Confidence:** 3

**Summary:**

This paper addresses the poor generalization of reward models in Reinforcement Learning from AI Feedback (RLAIF), which limits policy alignment due to distribution shift, label noise, and sample difficulty mismatches.
This paper proposes Curriculum-RLAIF, a framework that builds preference pairs of varying difficulty and trains the reward model with the curriculum from easy to difficult pairs.

**Strengths:**

1. This paper provides a novel aspect to address the reward model generalization issue (distribution shift, noise in labeling, capacity limit to learn hard data) in RLAIF by a data-centric approach.
2. This paper provides substantial experiments, including ablation studies, to demonstrate the effectiveness of its framework.

**Weaknesses:**

1. This method solves the reward model generalization issue at the cost of training different reward models for different policy models, respectively, which means the reward model and its curriculum data cannot be reused for different policy models. Authors should include this limitation in the paper.
2. Evaluation metrics are win scores that are judged by LLMs, which can introduce bias. Automatic benchmarks without LLMs are encouraged to be included, such as IFEval and LiveBench.
3. Presentation of this paper, especially the Experiments and Results sections, is confusing. Terminologies, concepts, and notations are not consistent. See Questions 1 to 5.
4. The motivation of this method is confusing. See Questions 6 and 7.

**Questions:**

1. What is the definition of $\Delta r$? Is $\Delta r$ defined differently in the Experiments than its first definition in Line 153? I'm assuming the reward in Line 153 is ground-truth reward and you use TextEval-Llama3.1-70B to estimate it. But in the Experiments section, you seem to use another way to estimate it?
2. Internal Eval, External Eval, and Implicit Eval are confusing for the first time. They sound like evaluation metrics. I suggest changing the name.
3. What is the formula for Internal Eval, External Eval, and Implicit Eval? It would be better to write them down clearly or attach them in the appendix.
4. For these three baselines, did you also divide them into 3 difficulty levels, easy, bridge, hard?
5. What is the initialization of the reward model and policy model in Table 1? Does the base model in this paper mean the instruct model instead of a pretrained model?
6. Can you elaborate on "However, relying on such estimates at scale is computationally expensive, as it requires reward evaluation across all query-response pairs."? Do you mean the number of pairs by ${n}\choose{2}$? But it seems you only need to run inference for $n$ times to obtain $r(y_i), i=1, ..., n$ and the computation of the difference is LLM-free, which is cheap.
7. A follow-up question for 6. "such estimates" is expensive, but auto-regressive sampling from the policy model is also expensive. I don't think the computational cost (FLOPs) of sampling from the policy model is cheaper than TextEval-Llama3.1-70B. If so, the motivation of the method is not valid.
8. Do the results suggest that Curriculum-RLAIF implicitly estimates $\Delta r$ better than Internal Eval?

I would like to raise the score to 6 if Question 6 and 7 can be addressed, and can further raise it to 8 if Question 1 to 5 can be resolved.

---

> ### Author Response · Authors · 2025-11-27
> **Response to Reviewer CQqb (1/3)**
>
> Many thanks to reviewer CQqb for insightful and detailed comments. The quality of the manuscript has been enhanced accordingly during the addressing of your comments and suggestions. We have responded to them one by one below, and hope your concerns are addressed.
>
> **W1:** The limitation discussion of the requirement of training different reward models when tasks change \
> **R1:** We agree there is a practical limitation that if the policy model class or data distribution changes substantially, the existing RM and its curriculum may no longer be optimal and may require additional adaptation or retraining. This limitation is shared by most RM-based RL/RLAIF pipelines [1-4], and our curriculum construction does not remove this dependency.
>
> Paper change: We will explicitly add this as a limitation in the paper (Limitations / Discussion).
>
> **W2:** Potential evaluation bias when using LLM as a judge \
> **R2:** We agree that LLM-as-a-judge metrics can introduce bias and should be interpreted with care. In our experiments, we follow common practice and use GPT-4o as a proxy judge to compare the policy output against the shared reference/SFT baseline, since it has been demonstrated in the literature that a strong LLM can work as a reliable proxy evaluator of humans (GPT-4 with over 80% agreement) [5].
> Due to its low cost, objectivity, and reproducibility, this evaluation method has been prevalently adopted in the literature of reward modeling and LLM alignment [2, 6].
>
> To mitigate the bias concern, we also include non-LLM-judge, label-based evaluations using human-annotated preference labels, and report reward score accuracy (i.e., how often the trained RM matches human preferences).
>
> Paper change: We will (i) explicitly discuss the potential bias of LLM-judging in the Evaluation/Limitations section, and (ii) move the human-label-based automatic metric (RM accuracy) to be more prominent in the main text (currently in Appendix E.1).
>
> **W3:** Experiments section presentation clarification (Q1-Q5)
>
> **Q1:**
> > What is the definition of $\Delta r$? Is $\Delta r$ defined differently in the Experiments than its first definition in Line 153? I'm assuming the reward in Line 153 is a ground-truth reward and you use TextEval-Llama3.1-70B to estimate it. But in the Experiments section, you seem to use another way to estimate it?
>
> **A1:** $\Delta r$ is defined in Line 153 as $\Delta r = | r(y_1) - r(y_2) |$, where $r({y_i})$ is the reward score predicted by the reward model given response $y_i$. $\Delta r$ is defined the same in the Experiments as in Line 153.
> Here, the reward model could be either an External Evaluator, e.g., TextEval-Llama3.1-70B in our experiments, or an Internal Evaluator, which is the reward model training online, or the Implicit Evaluation, which is the implicit reward model in DPO. This was briefly indicated in Line 267 and Line 278 in the experimental setup section.
>
> **Q2:**
> > Internal Eval, External Eval, and Implicit Eval are confusing for the first time. They sound like evaluation metrics. I suggest changing the name.
>
> **A2:** Thank you so much for this practical suggestion. We agree with you and we thought that changing the name to External Reward Estimation, Internal Reward Estimation, and Implicit Reward Estimation could be a more proper name, and could possibly mitigate the confusion regarding the definition of $\Delta r$ as you questioned earlier.
>
> **Q3:**
> > What is the formula for Internal Eval, External Eval, and Implicit Eval? It would be better to write them down clearly or attach them in the appendix.
>
> **A3:** We describe the terms between Line 267 and Line 278, and will further add formal formulations, as suggested by the reviewer, with indicators for different reward models in the appendix for clarity.
>
> **Q4:**
> > For these three baselines, did you also divide them into 3 difficulty levels, easy, bridge, hard?
>
> **A4:** These three curriculum baselines use the same collected dataset (for fairness) but do not rely on those discrete "contrastive → bridging → random" categories as we have designed in our proposed curricula.
> Instead, they estimate sample difficulty using the reward-distance score $\Delta r$
> and construct curricula by partitioning the dataset into four stages based on $\Delta r$ (i.e., difficulty is induced by $\Delta r$-based grouping rather than by our data-type design).

---

> ### Author Response · Authors · 2025-11-27
> **Response to Reviewer CQqb (2/3)**
>
> **Q5:**
> > What is the initialization of the reward model and policy model in Table 1? Does the base model in this paper mean the instruct model instead of a pretrained model?
>
> **A5:** Thanks for this in-depth technical question. This is critical to understand our experimental setups.
> The initialization of the policy model is the pretrained Gemma-1-2B or LLaMA-3-8B, as indicated by the “Base Model” column.  The initialization of the reward model is the pretrained Gemma-1-2B or LLaMA-3-8B, of which the unembedding layer is replaced with a randomly initialized 1-dimensional MLP output layer to produce a scalar value.
> This “pretrained backbone + new scalar head” initialization is common practice for reward modeling in prior work [1,2,3].
> In our paper, “base model” refers to the instruction-tuned pretrained model. This is also consistent with Figure 1, where inputs are formatted in a dialog style with role prefixes such as “Human” and “Assistant”.
>
> Paper change: we will make this wording explicit (and consistent) in the Table 1 caption and the surrounding experiment description to avoid ambiguity.
>
> **Q6:**
> > Can you elaborate on "However, relying on such estimates at scale is computationally expensive, as it requires reward evaluation across all query-response pairs."? Do you mean the number of pairs by? But it seems you only need to run inference for times to obtain and the computation of the difference is LLM-free, which is cheap.
>
> **A6:** The LLM inference for data generation of various methods is inevitable and the same. What we meant by "computationally expensive" is the additional reward-model inference required to obtain $r(y)$ in order to estimate the data difficulty, based on which the $\Delta r$-based curriculum baselines are constructed (e.g., sorting/partitioning into stages).
> By contrast, the proposed Curriculum-RLAIF does not require any reward-model inference for curriculum design (curriculum-design cost is 0 shown in Table 4), because difficulty is induced proactively by data construction rather than retroactively by scoring and sorting.
>
> We thank the reviewer for the careful check and will revise the sentence accordingly for clarity, e.g., "The curriculum baselines are created by sorting all pairs by $\Delta r$ and partitioning them into four stages (e.g., quartiles), from easy to hard, and training proceeds stage-by-stage. In contrast, Curriculum-RLAIF does not build stages by scoring/sorting all pairs with $\Delta r$, thus our curriculum design avoids the associated repeated reward-evaluation overhead."
>
> **Q7:**
> > A follow-up question for 6. "such estimates" is expensive, but auto-regressive sampling from the policy model is also expensive. I don't think the computational cost (FLOPs) of sampling from the policy model is cheaper than TextEval-Llama3.1-70B. If so, the motivation of the method is not valid.
>
> **A7:** To supplement A6, the autoregressive cost is not unique to our method. Any preference-data pipeline (including the baselines we compare to) needs to generate candidate responses from a policy/base model to form pairs; this sampling cost is therefore a shared “base cost.” Our computational-cost claim is about the additional overhead specific to $\Delta r$-based curriculum design.

---

> ### Author Response · Authors · 2025-11-27
> **Response to Reviewer CQqb (3/3)**
>
> **Q8:**
> > Do the results suggest that Curriculum-RLAIF implicitly estimates $\Delta r$ better than Internal Eval?
>
> **A8:** This is an insightful perspective. We **do not** make the claim that Curriculum-RLAIF “implicitly estimates $\Delta r$ better” than Internal Eval. Although Curriculum-RLAIF achieves comparable or better downstream policy performance than the Internal-Eval curriculum (Table 1), this does **not** directly imply superior difficulty estimation, because an effective curriculum does not necessarily correspond to a strict linear ordering of $\Delta r$, nor does it require sharp stage boundaries between stages.
>
> Instead, we interpret the empirical results as evidence about **curriculum design**. As discussed in Section 6:
>
> > As shown in Fig. 3 and Fig. 5, our curriculum at each stage spans a broader range of difficulty levels, yet achieves comparable or even superior performance compared to the internal-evaluation baseline. This suggests that overly strict selection based on estimated difficulty may not yield the most effective curriculum. Incorporating samples within a moderate difficulty range at each stage may act as an effective regularization that improves generalizability [7,8].
>
> Additionally, as pointed out by reviewer *59Mb*, the results may also be influenced by potential **forgetting** in Internal Eval when training on splits with clear distribution boundaries. We view this as an interesting research question and will explicitly highlight it as future work.
>
> Reference:
>
> [1] Ouyang et al., Training Language Models to Follow Instructions with Human Feedback, NeurIPS, 2022
>
> [2] Yang et al., RLCD: Reinforcement Learning from Contrastive Distillation for Language Model Alignment, ICLR 2024
>
> [3] Lee et al., RLAIF vs. RLHF: Scaling Reinforcement Learning from Human Feedback with AI Feedback, ICML 2024.
>
> [4] Rafailov et al., Direct Preference Optimization: Your Language Model is Secretly a Reward Model, NeurIPS, 2023
>
> [5] Zheng et al., Judging LLM-as-a-Judge with MT-Bench and Chatbot Arena, NeurIPS, 2023
>
> [6] Shaikh et al.,  Show, Don’t Tell: Aligning Language Models with Demonstrated Feedback, ICLR 2025
>
> [7] Srivastava et al., Dropout: A Simple Way to Prevent Neural Networks from Overfitting, JMLR. 2014.
>
> [8] Alex Hernández-García and Peter König, Data Augmentation Instead of Explicit Regularization, arXiv, 2020
>
> \
> We appreciate reviewer CQqb’s careful evaluation and constructive feedback. We hope our clarifications address the concerns raised and welcome any additional comments. We remain available to discuss further as needed.

---

### Official Review · Reviewer_eZBV · 2025-10-31

**Soundness:** 2
**Presentation:** 3
**Contribution:** 3
**Rating:** 6
**Confidence:** 3

**Summary:**

This paper proposes Curriculum-RLAIF, a framework for improving reward model generalizability in RLAIF through curriculum learning. Conventional reward models often suffer under distribution shifts, label noise, and varying sample difficulties. Their approach combines quality-aware sampling (guided prompting for easy samples, random sampling for hard samples) with controlled pairing to create preference data at varying difficulty levels. Reward models are trained progressively from easy contrastive pairs through intermediate "bridging pairs" to difficult random pairs. Experiments on harmlessness, helpfulness, and summarization tasks show consistent improvements over non-curriculum baselines across three tasks and two small-scale base models.

**Strengths:**

- Interesting preliminary study on the relationship between confidence of the reward model and the resulting reward distance.

- The proposed curriculum method is simple and data-centric, reducing reliance on both noisy preference labels and costly online difficulty scoring during training.

- In Tables 1-2, they show consistent improvements across three alignment tasks (harmlessness, helpfulness, summarization) and two base models (Gemma-1-2B, LLaMA-3-8B), suggesting the method generalizes across different settings.

- The paper systematically isolates contributions from data sources (Table 2: Drnd vs. curated mixture) and curriculum orderings (Table 3: Cbrg vs. Crev, Cdis, Cmix, Cach). This supports their design choices.

- In Figure 3 and Figure 5, the reward-distance distributions show that curriculum stages progress from easy to hard with smooth overlaps between adjacent stages. This seems to show that the curriculum generally works.

- The method requires expensive preference annotation (via off-the-shelf LLM) for only 25% of data (Drnd stage), while contrastive and bridging pairs are annotation-free. Table 4 shows a 75% reduction in labeling cost vs. conventional RLAIF.

**Weaknesses:**

**(1)** In Tables 1-2, they show that Curriculum-RLAIF outperforms strong baselines (Conventional RLAIF, Internal Eval) by 2-6 points. For example, on LLaMA-3-8B summarization, Conventional RLAIF achieves 0.84, Curriculum-RLAIF achieves 0.92. However, Internal Eval with curated data achieves 0.95. Without error bars or statistical tests, it is unclear if improvements are reliable. All results in Tables 1-5 lack error bars, confidence intervals, standard deviations, or multiple seeds. Given relatively small margins, it is not possible to assess significance. Bootstrap CIs over a larger number of test prompts or mean±std over multiple random seeds would help here.

**(2)** Limited task and model coverage:
- The authors only test dialog alignment (harmlessness/helpfulness) and summarization. No evaluation on reasoning tasks (math, code), instruction-following, or other domains where RLAIF is commonly applied.
- Only two small-scale backbones have been tested (2B, 8B parameters). No evaluation on larger models (30B+, 70B) or more recent reasoning-focused models (e.g., qwen-3, DeepSeek-R1-distill), where curriculum learning effects might differ. Broader model coverage would strengthen generalizability claims.

**(3)** The Method assumes that y+ ∼ p(y|x, g+) reliably beats y ∼ p(y|x) in D+brg and y ∼ p(y|x) reliably beats y− ∼ p(y|x, g−) in D−brg. However, there is no empirical validation of:
- How often does guided-good fail to beat random?
- How often does random fail to beat guided-bad?
- How is violated preference noise handled?
- Table 3 shows anchored curriculum Cach (which enforces preference via conditioning) achieves comparable performance, suggesting these assumptions may not be critical.

**(4)** All policy evaluations use GPT-4o as a judge (Sec. 4), and the robustness to alternative judges has not been unexplored. I don’t think this is a major problem. But it would be interesting to see an exploration of different judges here, who may differ in their preferences.

**(5)** In Table 4, Appendix D, the authors only count the preference annotation cost, but ignore the response generation overhead. The guided sampling requires additional prompting (contrastive instructions). Moreover, no wall-clock times or FLOPs counts are reported. The Internal Eval may be more expensive than claimed due to repeated evaluations (9/4 N factor), but an actual runtime comparison is missing.

**Questions:**

1. Can you provide error bars or confidence intervals for all results? Specifically, bootstrap 95% CIs over the 1000 test prompts, or mean±std over random seeds for Tables 1-3 and Table 5? Which improvements remain statistically significant after accounting for variance?

2. For Drnd pairs, what percentage uses LLM labeling vs. human annotation? What are the inter-annotator agreement stats? What quality filters or checks were applied to CoT reasoning chains from LLaMA-3.3-70B? Any calibration analysis comparing LLM judgments to human judgments?

3. How sensitive are results to alternative judges? Can you provide cross-judge agreement analysis on a subset?

4. How does the approach perform on larger models (30B+, 70B parameters) or recent reasoning models? Moreover, does your method have value for other reasoning tasks (math, code) beyond dialog alignment and summarization?

6. What are the actual wall-clock times and total FLOPs for all methods, including response generation? How does guided sampling overhead compare to preference labeling cost in practice?

---

> ### Author Response · Authors · 2025-11-27
> **Response to Reviewer eZBV (1/3)**
>
> We thank reviewer eZBV for recognizing the contributions of our work and for the insightful comments that help enhance its quality. We sincerely appreciate the feedback. Below, we provide responses to each comment and hope they address the concerns.
>
> **W1:** Error bars or statistical tests in reported experimental results \
> **R1:** Thanks for the comment and suggestion. We fully agree that incorporating error bars or statistical tests would help to assess the significance reliably. Following the suggestion, we ran additional experiments with five random seeds and updated Tables 1 and 2 accordingly below. Each entry now reports the mean and standard deviation.
>
> **Table 1**
> | **Base Model** |  **Category**  | **Method**   | **Harmlessness** | **Helpfulness** | **Summary** |
> |----------------|-----| --------------------|------------------|-----------------|-------------|
> | Gemma-1-2B  | Non-curriculum   | CAI    | 0.79 $\pm$ 0.01  | 0.85  $\pm$ 0.02   | 0.75 $\pm$ 0.01   |
> | Gemma-1-2B  |  Non-curriculum | RLCD   | 0.83  $\pm$ 0.02   | 0.87 $\pm$ 0.03   | 0.77 $\pm$ 0.02   |
> | Gemma-1-2B  |  Non-curriculum | Conventional RLAIF      | 0.83 $\pm$ 0.04 | 0.86 $\pm$ 0.02  | 0.80 $\pm$ 0.01 |
> | Gemma-1-2B  | Curriculum  | Internal Eval.   | 0.90  $\pm$ 0.02    | 0.88 $\pm$ 0.02   | 0.85 $\pm$ 0.02  |
> | Gemma-1-2B  | Curriculum  | External Eval.    |  0.88 $\pm$ 0.03    | 0.87 $\pm$ 0.02  | 0.86 $\pm$ 0.01 |
> | Gemma-1-2B  | Curriculum  | Implicit Eval. (DPO)    | 0.86 $\pm$ 0.03  |  0.85 $\pm$ 0.01 | 0.82 $\pm$ 0.02 |
> | Gemma-1-2B  | Curriculum  | Curriculum-RLAIF    | **0.92 $\pm$ 0.02**  | **0.93$\pm$ 0.01**  | **0.88$\pm$ 0.01**  |
> ||
> | LLaMA-3-8B  | Non-curriculum   | CAI   | 0.83 $\pm$ 0.02  | 0.87  $\pm$ 0.02 | 0.79 $\pm$ 0.01  |
> | LLaMA-3-8B |  Non-curriculum | RLCD   | 0.85  $\pm$ 0.01   | 0.88 $\pm$ 0.02   | 0.81 $\pm$ 0.02   |
> | LLaMA-3-8B |  Non-curriculum | Conventional RLAIF      | 0.88 $\pm$ 0.03             | 0.90 $\pm$ 0.04            | 0.85 $\pm$ 0.02        |
> | LLaMA-3-8B| Curriculum  | Internal Eval.     | 0.89  $\pm$ 0.02          | 0.91 $\pm$ 0.02   | 0.90 $\pm$ 0.01      |
> | LLaMA-3-8B | Curriculum  | External Eval.         |  0.85 $\pm$ 0.02    | 0.87 $\pm$ 0.03     | 0.89 $\pm$ 0.03        |
> | LLaMA-3-8B  | Curriculum  | Implicit Eval. (DPO)      | 0.90 $\pm$ 0.01             |  0.90 $\pm$ 0.02          | 0.87 $\pm$ 0.02       |
> | LLaMA-3-8B  | Curriculum  | Curriculum-RLAIF        | **0.93 $\pm$ 0.03**             | **0.95$\pm$ 0.02**            | **0.92$\pm$ 0.01**        |
>
> **Table 2**
> | **Base Model** |  **Data Source**  | **Method**   | **Harmlessness** | **Helpfulness** | **Summary** |
> |----------------|-----| --------------------|------------------|-----------------|-------------|
> | Gemma-1-2B  | $D_{rnd}$  | Internal Eval.    | 0.90 $\pm$ 0.02    | 0.88  $\pm$ 0.02          | 0.85 $\pm$ 0.02        |
> | Gemma-1-2B  |  $D_{rnd}$  | External Eval.  | 0.88  $\pm$ 0.03    | 0.87 $\pm$ 0.02            | 0.86 $\pm$ 0.01        |
> | Gemma-1-2B  |  $D_{rnd}$  | Implicit Eval. (DPO)       | 0.86 $\pm$ 0.03             | 0.85 $\pm$ 0.01            | 0.82 $\pm$ 0.02        |
> | Gemma-1-2B  | $D_{ctr} + D_{brg}^{+-} + D_{rnd}$  | Internal Eval.  | **0.93  $\pm$ 0.01**          | 0.91 $\pm$ 0.02          | 0.88 $\pm$ 0.01      |
> | Gemma-1-2B  | $D_{ctr} + D_{brg}^{+-} + D_{rnd}$  | External Eval. |  0.91 $\pm$ 0.03    | 0.90 $\pm$ 0.01           | **0.89 $\pm$ 0.02**       |
> | Gemma-1-2B  | $D_{ctr} + D_{brg}^{+-} + D_{rnd}$  | Implicit Eval. (DPO)      | 0.90 $\pm$ 0.01             |  0.88 $\pm$ 0.04           | 0.87 $\pm$ 0.02       |
> | Gemma-1-2B  | $D_{ctr} + D_{brg}^{+-} + D_{rnd}$  | Curriculum-RLAIF   | 0.92 $\pm$ 0.02    | **0.93$\pm$ 0.01**  | 0.88$\pm$ 0.01   |
> ||
> | Gemma-1-2B  | $D_{rnd}$  | Internal Eval.   | 0.89 $\pm$ 0.02 | 0.91  $\pm$ 0.02  | 0.90 $\pm$ 0.01  |
> | Gemma-1-2B  |  $D_{rnd}$  | External Eval.  | 0.85  $\pm$ 0.02 | 0.87 $\pm$ 0.03  | 0.89 $\pm$ 0.03   |
> | Gemma-1-2B  |  $D_{rnd}$  | Implicit Eval. (DPO)       | 0.90 $\pm$ 0.01             | 0.90 $\pm$ 0.02   | 0.87 $\pm$ 0.02 |
> | Gemma-1-2B  | $D_{ctr} + D_{brg}^{+-} + D_{rnd}$  | Internal Eval.     | 0.91  $\pm$ 0.02          | 0.93 $\pm$ 0.02          | **0.95 $\pm$ 0.02**   |
> | Gemma-1-2B  | $D_{ctr} + D_{brg}^{+-} + D_{rnd}$  | External Eval.  |  0.88 $\pm$ 0.03    | 0.91 $\pm$ 0.03    | 0.91 $\pm$ 0.01        |
> | Gemma-1-2B  | $D_{ctr} + D_{brg}^{+-} + D_{rnd}$  | Implicit Eval. (DPO)   | 0.92 $\pm$ 0.02   |  0.91 $\pm$ 0.03  | 0.89 $\pm$ 0.03  |
> | Gemma-1-2B  | $D_{ctr} + D_{brg}^{+-} + D_{rnd}$  | Curriculum-RLAIF   | **0.93 $\pm$ 0.03**   | **0.95$\pm$ 0.02**  | 0.92$\pm$ 0.01  |
>
> The updated experimental results are consistent with our original single-seed findings and do not challenge the conclusions presented in the manuscript. The small standard deviations indicate that the models and methods are robust to random seed variation in alignment tasks, which aligns with prior results reported in [1].

---

> ### Author Response · Authors · 2025-11-27
> **Response to Reviewer eZBV (2/3)**
>
> **W2:** Limited task and model coverage \
> **R2:** We thank the reviewer for suggesting more comprehensive experimental setups to enhance the empirical evaluation of our method. We agree that testing on instruction-following tasks would substantially improve the evidence for effectiveness and generalizability. In contrast, we believe evaluating on reasoning tasks (e.g., code generation and mathematical problem-solving) is less aligned with the scope of this work. To the best of our knowledge, RL-based reasoning enhancement typically follows a different pipeline from LLM alignment, particularly in reward modeling: code and math tasks can leverage rule-based rewards, whereas the common alignment tasks we study require training neural network–based reward models. Since our contribution focuses on the generalizability of reward-model training, math and code generation are not the most appropriate domains for evaluation. We are actively adding experiments on instruction-following benchmarks (e.g., AlpacaEval) and aim to finalize these results before the end of the discussion period.
>
> We appreciate the suggestion to evaluate larger base models to strengthen the generalizability claims of our method. As discussed in the paper (Appendix, Table 4), curriculum-based approaches introduce additional complexity for multi-stage reward model training and incur substantial compute overhead in both data construction and curriculum design. Running full experiments at the 70B scale would therefore be prohibitively costly within the rebuttal window. We agree that this is an important direction and plan to pursue it in follow-up work. In the meantime, we are actively running experiments with a 30B+ model and will update the rebuttal with results as soon as they complete.
>
> **W3:** statistics regarding the noise in the constructed dataset. \
> **R3:** Thank you for this question to help us gain a more in-depth view of the automatically constructed dataset. We are working on these experiments as well and will update the results once the experiments complete.
>
> **W3.4:**
> > Table 3 shows anchored curriculum Cach (which enforces preference via conditioning) achieves comparable performance, suggesting these assumptions may not be critical.
>
> **R3.4:** The anchored curriculum method adopts a different method to construct preference dataset and also induces possible preference label noise in the dataset.
> Specifically, we posit that label noise depends on the base model’s conditional compatibility: our method conditions only on a guidance prompt, whereas the anchored curriculum conditions on both an anchor response and a guidance prompt.
> From Table 3, the anchored curriculum delivers comparable but slightly weaker performance than the bridging curriculum. We hypothesize that the anchored approach induces more noise because its conditional generation task is more complex and demanding. Viewed through this lens, the assumption remains central to our method.
>
> **W4:** other LLMs as a judge besides GPT-4o \
> **R4:** Thank you for this comment. We agree that LLM-as-a-judge metrics can introduce bias and should be interpreted with care. In our experiments, we follow common practice and use GPT-4o as a proxy judge to compare the policy output against the shared reference/SFT baseline, since it has been demonstrated in the literature that a strong LLM can work as a reliable proxy evaluator of humans (GPT-4 with over 80% agreement) [4].  Due to its low cost, objectivity, and reproducibility, this evaluation method has been prevalently adopted in the literature of reward modeling and LLM alignment [2, 5].
>
> To mitigate the bias concern, we also include non-LLM-judge, label-based evaluations using human-annotated preference labels, and report reward score accuracy (i.e., how often the trained RM matches human preferences) in Figure 5 in the Appendix.

---

> ### Author Response · Authors · 2025-11-27
> **Response to Reviewer eZBV (3/3)**
>
> **W5**:
> > In Table 4, Appendix D, the authors only count the preference annotation cost, but ignore the response generation overhead. The guided sampling requires additional prompting (contrastive instructions). Moreover, no wall-clock times or FLOPs counts are reported. The Internal Eval may be more expensive than claimed due to repeated evaluations (9/4 N factor), but an actual runtime comparison is missing.
>
> **R5:** Thanks for the thoughtful comment.  In Table 4, the “Data Construction” column reports response-generation overhead as a function of the average sequence length $L$. We exclude additional tokens from the guidance prompts because they are very short, as shown in Figure 1 prompts like “please generate a longer summary” comprise only a few tokens.
> Relative to source inputs that often span hundreds of tokens (e.g., articles in the summarization task), this prompt overhead is negligible and does not materially affect the compute estimates.
>
> The Internal Eval is expensive due to repeated evaluation. We evaluate the wall-clock times of the internal evaluation process for curriculum design as below:
> - Dataset: Harmlessness dataset
> - Base model: LLaMA3-8B
> - GPUs: 4 x NVIDIA A100 (80G) GPUs
> - Wall-clock time to evaluate the dataset once: about 87 minutes
>
> **Q1:**
> > Can you provide error bars or confidence intervals for all results? Specifically, bootstrap 95% CIs over the 1000 test prompts, or mean±std over random seeds for Tables 1-3 and Table 5? Which improvements remain statistically significant after accounting for variance?
>
> **A1:** Please see our additional results and discussion in the response to W1.
>
> **Q2:**
> > For $D_{rnd}$ pairs, what percentage uses LLM labeling vs. human annotation? What are the inter-annotator agreement stats? What quality filters or checks were applied to CoT reasoning chains from LLaMA-3.3-70B? Any calibration analysis comparing LLM judgments to human judgments?
>
> **A2:** All the response pairs in $D_{rnd}$  are annotated using LLM labeling, specifically using LLaMA-3.3-70B. This was introduced between Line 282 to 284. We didn’t apply quality filtering techniques in our annotation procedures. We adopted techniques proposed in paper [3] to incorporate CoT to improve the quality of LLM annotation. Specifically, this method employs a two-step prompting approach for preference labeling. The first step is to generate rationales, and the second step is to produce preference labels based on the generated rationales. Please see prompt examples for different tasks in Appendix H for more details.
> The practical technique is from prior work [3], where in-depth analysis comparing LLM judgments to human judgments was conducted.
>
> **Q3:**
> > How sensitive are results to alternative judges? Can you provide cross-judge agreement analysis on a subset?
>
> **A3:** Please see our response to W4.
>
> **Q4:**
> > How does the approach perform on larger models (30B+, 70B parameters) or recent reasoning models? Moreover, does your method have value for other reasoning tasks (math, code) beyond dialog alignment and summarization?
>
> **A4:** Please see our response to W2.
>
> **Q5:**
> > What are the actual wall-clock times and total FLOPs for all methods, including response generation? How does guided sampling overhead compare to preference labeling cost in practice?
>
> **A5:** We measured wall-clock time for internal evaluations and analyzed guided sampling overhead in our response to W5. Due to the limited rebuttal window, we were unable to complete wall-clock measurements for all methods. We will provide a comprehensive set of timing results in the final version.
>
> Reference
>
> [1] Ouyang et al., Training Language Models to Follow Instructions with Human Feedback, NeurIPS 2022 \
> [2] Yang et al., RLCD: Reinforcement Learning from Contrastive Distillation for Language Model Alignment, ICLR 2024 \
> [3] Lee et al., RLAIF vs. RLHF: Scaling Reinforcement Learning from Human Feedback with AI Feedback, ICML 2024 \
> [4] Zheng et al., Judging LLM-as-a-Judge with MT-Bench and Chatbot Arena, NeurIPS 2023 \
> [5] Shaikh et al.,  Show, Don’t Tell: Aligning Language Models with Demonstrated Feedback, ICLR 2025
>
> We thank reviewer eZBV for the careful evaluation and constructive feedback. We hope our clarifications address the concerns raised and welcome any further comments.

---

> ### Author Response · Authors · 2025-12-01
> **Response to to Reviewer eZBV (Updated Experimental Results 1/2)**
>
> **Additional experimental results to W2 and Q4**
>
> Thank you for suggesting an evaluation on larger-scale models to strengthen our generalizability claims. Below, we present additional results using Qwen2.5-32B, a widely used model in the 30B+ pretrained LLM category:
>
> **Table: Comparison between policy performance trained through our method and various baselines. This table will be integrated into Table 1.**
> | **Base Model** |  **Category**  | **Method**   | **Harmlessness** | **Helpfulness** | **Summary** |
> |----------------|-----| --------------------|------------------|-----------------|-------------|
> | Qwen2.5-32B  | Non-curriculum   | CAI    | 0.88 $\pm$ 0.01  | 0.89  $\pm$ 0.01   | 0.86 $\pm$ 0.01   |
> | Qwen2.5-32B  |  Non-curriculum | RLCD   | 0.89  $\pm$ 0.01   | 0.92 $\pm$ 0.02   | 0.87 $\pm$ 0.01   |
> | Qwen2.5-32B  |  Non-curriculum | Conventional RLAIF      | 0.91 $\pm$ 0.02 | 0.93 $\pm$ 0.03  | 0.90 $\pm$ 0.02 |
> | Qwen2.5-32B  | Curriculum  | Internal Eval.   | 0.93  $\pm$ 0.01    | 0.94 $\pm$ 0.01   | 0.92 $\pm$ 0.02  |
> | Qwen2.5-32B  | Curriculum  | External Eval.    |  0.90 $\pm$ 0.03    | 0.91 $\pm$ 0.02  | 0.93 $\pm$ 0.01 |
> | Qwen2.5-32B  | Curriculum  | Implicit Eval. (DPO)    | 0.94 $\pm$ 0.01  |  0.93 $\pm$ 0.01 | 0.91 $\pm$ 0.03 |
> | Qwen2.5-32B  | Curriculum  | Curriculum-RLAIF    | **0.96 $\pm$ 0.01**  | **0.97$\pm$ 0.01**  | **0.95$\pm$ 0.02**  |
>
>
> **Table: Comparison between policy performance trained through various curriculum methods using different data sources. This table will be integrated into Table 2.**
> | **Base Model** |  **Data Source**  | **Method**   | **Harmlessness** | **Helpfulness** | **Summary** |
> |----------------|-----| --------------------|------------------|-----------------|-------------|
> | Qwen2.5-32B  | $D_{rnd}$  | Internal Eval.    | 0.93 $\pm$ 0.01   | 0.94  $\pm$ 0.01          | 0.92 $\pm$ 0.02        |
> | Qwen2.5-32B  |  $D_{rnd}$  | External Eval.  | 0.90  $\pm$ 0.03    | 0.91 $\pm$ 0.02            | 0.93 $\pm$ 0.01        |
> | Qwen2.5-32B |  $D_{rnd}$  | Implicit Eval. (DPO)       | 0.94 $\pm$ 0.01             | 0.93 $\pm$ 0.01            | 0.91 $\pm$ 0.03        |
> | Qwen2.5-32B  | $D_{ctr} + D_{brg}^{+-} + D_{rnd}$  | Internal Eval.  | 0.94  $\pm$ 0.02          | 0.96 $\pm$ 0.02          | 0.94 $\pm$ 0.02      |
> | Qwen2.5-32B | $D_{ctr} + D_{brg}^{+-} + D_{rnd}$  | External Eval. |  0.93 $\pm$ 0.03    | 0.93 $\pm$ 0.01           | 0.93 $\pm$ 0.02       |
> | Qwen2.5-32B | $D_{ctr} + D_{brg}^{+-} + D_{rnd}$  | Implicit Eval. (DPO)      | 0.95 $\pm$ 0.01             |  0.95 $\pm$ 0.01           | 0.93 $\pm$ 0.01       |
> | Qwen2.5-32B | $D_{ctr} + D_{brg}^{+-} + D_{rnd}$  | Curriculum-RLAIF   | **0.96 $\pm$ 0.01**    | **0.97$\pm$ 0.01**  | **0.95$\pm$ 0.02**   |
>
> The additional results indicate that our method generalizes well across model scales from 2B to 32B. Performance improves monotonically with model size, and our Curriculum-RLAIF consistently enhances alignment at every scale. We hope these additional experiments, together with the results reported in the paper, address your concern about how model scale affects the effectiveness of our method.

---

> ### Author Response · Authors · 2025-12-01
> **Response to to Reviewer eZBV (Updated Experimental Results 2/2)**
>
> **Additional experimental results to W3**
>
> We empirically validated the preference assumption within the bridging pairs $\mathcal{D}_\text{brg}$ in our constructed dataset. Specifically, we measured win/tie/loss rates for (i) the positively guided response ($y^+$) versus the randomly sampled response  ($y$), and (ii) the randomly sampled response ($y$) versus the negatively guided response ($y^-$), using GPT-4 as the judge. The experimental results are provided below. Entries in the $\textit{Loss}$ column indicate the "failure rate", i.e., cases where the assumptions $y^+ \succ y$ or $y \succ y^-$ are violated.
>
>
> | **Base Model** |  **Dataset**  | **Comparison Pair**   | **Win (%)** | **Tie (%)** | **Loss (Failure) (%)** |
> |----------------|-----| --------------------|------------------|-----------------|-------------|
> | Gemma-1-2B  | Harmlessness | $y^+$ vs. $y$   | 74.2  | 13.4         | 12.4     |
> | Gemma-1-2B  | Harmlessness | $y$ vs. $y^-$    | 78.5  | 10.1         | 11.4     |
> | Gemma-1-2B  |  Helpfulness | $y^+$ vs. $y$  | 75.8   | 8.8           | 15.4       |
> | Gemma-1-2B  |  Helpfulness | $y$ vs. $y^-$  | 79.1   | 7.1           | 13.8        |
> | Gemma-1-2B |  Summary  | $y^+$ vs. $y$      | 72.5           | 14.2         | 13.3       |
> | Gemma-1-2B |  Summary  | $y$ vs. $y^-$    | 75.4           | 12.5           | 12.1    |
> ||
> | LLaMA-3-8B   | Harmlessness | $y^+$ vs. $y$   | 87.8 | 6.1     | 6.1    |
> | LLaMA-3-8B  | Harmlessness | $y$ vs. $y^-$    | 91.5  | 4.2         | 4.3     |
> | LLaMA-3-8B  |  Helpfulness | $y^+$ vs. $y$  | 85.4   | 5.8           | 8.8      |
> | LLaMA-3-8B   |  Helpfulness | $y$ vs. $y^-$  | 89.1   | 3.7         | 7.2       |
> | LLaMA-3-8B  |  Summary  | $y^+$ vs. $y$      | 83.4          | 8.8        | 7.8     |
> | LLaMA-3-8B |  Summary  | $y$ vs. $y^-$    | 86.2           | 7.1          | 6.7   |
> ||
> | Qwen2.5-32B    | Harmlessness | $y^+$ vs. $y$   | 96.1  | 2.2         | 1.7     |
> | Qwen2.5-32B  | Harmlessness | $y$ vs. $y^-$    | 97.5 | 1.3         | 1.2     |
> | Qwen2.5-32B   |  Helpfulness | $y^+$ vs. $y$  | 93.8 | 2.8           | 3.4       |
> | Qwen2.5-32B   |  Helpfulness | $y$ vs. $y^-$  | 95.9   | 1.8           | 2.3        |
> | Qwen2.5-32B   |  Summary  | $y^+$ vs. $y$      | 92.5           | 4.4         | 3.1       |
> | Qwen2.5-32B  |  Summary  | $y$ vs. $y^-$    | 94.8           | 3.1           | 2.1    |
>
> The results indicate that even relatively small LLMs can be effectively steered using our guided sampling method. We can also observe a clear trend that as model size increases, failure rates decrease. This is reasonable as larger base models exhibit stronger instruction-following capabilities, enabling guided sampling to align more accurately with the intended guidance.

---

### Official Review · Reviewer_SkUA · 2025-11-01

**Soundness:** 3
**Presentation:** 3
**Contribution:** 2
**Rating:** 4
**Confidence:** 5

**Summary:**

The paper addresses a key limitation of Reinforcement Learning from AI Feedback (RLAIF) - the poor generalization of reward models, which negatively impacts downstream policy alignment.

To mitigate this issue, the authors propose Curriculum-RLAIF, a static curriculum strategy comprising three stages:
- Sampling completions for prompts using a mixture of guided (Yang et al., 2024) and random sampling,
- Pairing completions to form a preference dataset, and
- Applying an easy-to-hard curriculum over the preference dataset using coarse-grained stages.

Empirical results demonstrate that Curriculum-RLAIF improves policy alignment performance over both non-curriculum and curriculum-based baselines.

**Strengths:**

The paper is clearly written, with a well-motivated discussion of existing RLAIF limitations and a comprehensive review of related work.

The proposed approach is conceptually simple yet effective, showing consistent improvements across experimental settings.

The empirical analysis is systematic, supported by detailed ablations and comparisons.

**Weaknesses:**

The proposed Curriculum-RLAIF is a static curriculum strategy, while the internal evaluation baseline used in experiments is adaptive to the reward model's learning progress.

Table 2 suggests that the internal evaluation baseline could perform even better if applied to the full dataset (not just $\mathcal{D}_\text{rnd}$) and with more fine-grained curriculum stages. While adaptive strategies incur higher computational cost, they are likely to yield better final performance than static curricula.

It would be valuable to compare the proposed method and the adaptive baseline (allowing more fine-grained curriculum stages for the baseline) in terms of final alignment performance and total wall-clock time.

Additional clarifications and considerations (no new experiments needed):
- Constructing a full preference dataset $\mathcal{D}_\text{full}$, including all pairs of completions (contrastive, bridge, random) per prompt, may better capture prompt-level difficulty.
- The proposed approach uses an easy-to-hard ordering; however, examples of moderate difficulty can sometimes yield better learning efficiency.
- An adaptive variant could periodically sample prompts and select completion pairs with moderate reward gaps based on the current reward model (internal evaluation).

**Questions:**

Please see the discussions in the weaknesses section above.

In Table 1, it appears that curriculum baselines use only $\mathcal{D}_\text{rnd}$, while the proposed Curriculum-RLAIF uses the full dataset. In Table 2, baselines seem more competitive with the full dataset. Please clarify this explicitly in the table caption.

For both Curriculum-RLAIF and the curriculum baselines, how is the stage transition determined? Is it based on training for a fixed number of epochs per stage before moving to the next?

---

> ### Author Response · Authors · 2025-11-27
> **Response to Reviewer SkUA (1/2)**
>
> We thank reviewer SkUA for the insightful and constructive feedback, which substantially improved the quality of the manuscript. We also appreciate the reviewer for the encouraging comments on the merit of our proposed method and thoroughness of our empirical analysis and ablations. Below, we address each concern and question in turn. We hope these clarifications resolve the earlier points raised.
>
> **W1:** Potential better performance of the internal evaluation method when using more finer-grained curriculum stages. \
> **R1:** Thanks for this insightful question, which touches key characteristics of our method: it is simple yet effective, and it is static with respect to the number of curriculum stages. We agree that it would be interesting to explore whether finer-grained staging could further leverage the internal evaluation approach.
>
> Following your suggestion, we are running additional experiments that doubled the number of curriculum stages to eight for the internal evaluation baseline and measured the associated computational cost. We would like to note that the the strong internal evaluation method in Table 2 already benefits from the constructed preference data by our Curriculum-RLAIF pipeline, i.e, $D_{ctr} + D_{brg}^{+/-} + D_{rnd}$. We retain this setup in the new experiment to test whether further gains are possible on top of this strong method. We will report the experimental results in the rebuttal phase once the experiments complete.
>
> **W2:** Constructing a full preference dataset, including all pairs of completions per prompt, may better capture prompt-level difficulty. \
> **R2:** We appreciate the question and fully agree with the underlying idea. In the current manuscript, we construct a non-overlapping dataset for curriculum methods to enable *a fair comparison with existing non-curriculum baselines*. Concretely, we partition the original dataset for curriculum construction rather than performing multi-round data generation. This non-overlapping setup ensures that every method in the comparison sees each prompt only once per epoch. At the same time, our approach inherently has the capacity to expand the dataset, thanks to its fully automatic data curation pipeline.
>
> **W3:** The proposed approach uses an easy-to-hard ordering; however, examples of moderate difficulty can sometimes yield better learning efficiency. \
> **R3:** We strongly agree with this point and have explicitly accounted for the learning dynamics in our design by introducing *bridging pairs*. These pairs both smooth the difficulty gap between stages and inject examples of moderate difficulty.
>
> An interesting finding that is relevant to this idea is the analysis presented in Figure 3 and Figure 5, where there are "difficult overlaps” between data of adjoint stages. This suggests that some more difficult cases could “invade” the data split of the current stage. We attribute this to the stochasticity of LLM generation, and view it as beneficial: the controlled spillover regularizes stage boundaries and supports more robust learning, consistent with our discussion in the Response to W1 and Section 6.
>
> **W4:**  An adaptive variant could periodically sample prompts and select completion pairs with moderate reward gaps based on the current reward model (internal evaluation). \
> **R4:** Thank you for the thoughtful suggestion. It aligns with your earlier hypothesis about the value of moderately difficult examples. Given the current manuscript’s scope, we plan to treat this as a promising extension, alongside the ***hybrid variant*** outlined in Section 6:
> > Our experiments demonstrated that curriculum design using the internal reward model itself is an effective approach. It offers the advantage of finer granularity in curriculum construction, which has the potential to further improve performance, however, at the cost of exponentially increasing computational costs. Exploring *hybrid approaches* that combine the strengths of our pre-hoc distribution-bridging method with online internal evaluation methods would be a valuable direction for future research. For example, performing online evaluation and data selection within a small-scale subset pre-constructed by our method could lead to a balance between improved performance and reduced computational costs.
>
> Beyond these specific directions, we see substantial room for further study both in technical practices and foundational research. In particular, understanding the regularization role of moderate cases remains an open avenue. We believe that our comprehensive analysis and the flexible framework introduced in this work provide a solid basis for exploring these directions.

---

> ### Author Response · Authors · 2025-11-27
> **Response to Reviewer SkUA (2/2)**
>
> **Q1:** In Table 1, it appears that curriculum baselines use only $D_{rnd}$, while the proposed Curriculum-RLAIF uses the full dataset. In Table 2, baselines seem more competitive with the full dataset. \
> **A1:** Thanks for pointing out the confusion. We will revise the manuscript accordingly to address this potential source of confusion. To clarify: the dataset $D_{rnd}$ is actually the data source used in prior work [1,2,3], whereas the “full data”, i.e., $D_{ctr} + D_{brg}^{+/-} + D_{rnd}$, is the curated set we constructed via the Curriculum-RLAIF data pipeline.
> We conducted experiments with either naive random data $D_{rnd}$ and intentional difficulty controlled data, $D_{ctr} + D_{brg}^{+/-} + D_{rnd}$, as an ablation to investigate the impact of the data sources in the task of reward modeling and LLM alignment.
> This study demonstrates that simply adopting the data source, i.e., $D_{ctr} + D_{brg}^{+/-} + D_{rnd}$, constructed by Curriculum-RLAIF can boost existing methods by a large margin.
>
> References \
> [1] Bai et al., Constitutional AI: Harmlessness from AI feedback, arXiv, 2022 \
> [2] Grattafiori et al., The Llama 3 Herd of Models, arXiv, 2024 \
> [3] Lee et al., RLAIF vs. RLHF: Scaling Reinforcement Learning  from Human Feedback with AI Feedback, ICML, 2024
>
> We thank reviewer SkUA again for the careful evaluation and constructive recommendations. We hope the clarifications provided address the concerns and remain available for further correspondence.

---

### Official Review · Reviewer_59Mb · 2025-11-03

**Soundness:** 1
**Presentation:** 2
**Contribution:** 2
**Rating:** 2
**Confidence:** 3

**Summary:**

This paper tackles  the problem of training reward models with curriculum to improve generalization. First, the authors expose the fundamental challenges of learning from diverse quality data i.e. evaluating the difficulty of a pair of responses and show that reward differences from a pre-trained LLM across the responses are surrogates for label noise / difficulty. Further, using controlled data generation and prompt augmentation they generate a dataset of increasingly difficult preference pairs. Training with the introduced curriculum leads to better reward models and better policy performance after PPO finetuning. over baselines without a curriculum. Further, they ablate their curriculum design to show the contribution of the different components.

**Strengths:**

- The paper presents a thorough literature review and the problem statement is clearly laid out.
- It presents an interesting analysis on an existing dataset where the response pairs with low confidence scores (i.e. humans find them difficult to label) also exhibited lower accuracy when using an off-the shelf LLM for ratings.
- The authors further show that existing models have a higher difference in predicted rewards which enables them to use this as a surrogate to evaluate response pair noise or difficulty.
- The method implicitly uses the way responses are generated to pair them up and create a curriculum of easy to difficult response pairs, which avoids the expensive LLM based evaluations.
- The authors carefully ablate different parts of the method, including the dataset sources and curriculum design. However, the ablation presents mixed results towards the utility of the proposed method (while it does outperform all the non-curriculum based methods).

**Weaknesses:**

- Intuitively it seems like since y1 and y2 are labelled using human annotations or a better model with context, they should have lower noise than pairing a bad sample with a random sample as the latter case uses less context to generate the preference ranking? So, this seems to be a wrong preference order? It would be interesting to evaluate the validity of the curriculum design against a stronger model or human evals.
- Table 2 shows that the proposed curriculum is not the dominant method and the base evaluation methods outperform Curriculum-RLAIF. Also the performance for the baseline curriculum methods in Table 1 is lower than the ones in Table 2? What is the data source for the results in the Table 1? And ideally it should be the best performing setting across each baseline that is presented in a single table.
- Overall, the paper is well written but the experiments do not provide strong evidence that the proposed curriculum method significantly outperforms the baselines specially the internal evaluation baseline if the preference data is a mixing of the paired and random pairs. If the authors can provide additional evidence, and clarify the above questions, I will be happy to raise my scores.
- Typo in table 1
- Looking at Figure 3, it seems like the Internal evaluation method creates distinct non-overlapping chunks of the dataset and should be prone to catastrophic forgetting but that does not seem to be happening. Is the curriculum training merging the new dataset?

**Questions:**

See weaknesses above.

---

> ### Author Response · Authors · 2025-11-27
> **Response to Reviewer 59Mb (1/2)**
>
> We thank reviewer 59Mb very much for the careful reading and for providing constructive feedback that has contributed to enhancing the quality of this paper. We address each concern in turn, following the numbering of the review.
>
> **W1:** potential wrong preference order issue and evaluation of curriculum design against a stronger model or human evals.
>
> **R1:** Thanks for raising this point and highlighting a common source of confusion. In canonical RLHF and RLAIF methods, response pairs $\\{ y_1, y_2 \\}$ are drawn from randomly sampled generations. The key difference lies in how preferences are annotated: RLHF uses human annotators, whereas RLAIF relies on an LLM as a proxy annotator.
>
> A central challenge we address is the "wrong preference order" issue as you suggested, which we refer to as “noisy labels” in the paper. We hypothesized that our curriculum training strategy would make the reward model more robust to such noise than conventional approaches. Treating *human annotations as ground truth*, we quantitatively evaluated the error rate of preference labels produced by different methods. This analysis appears in Appendix E.1 (Table 5 and Figure 4), and we present the results below for convenience:
> | **Base Model** | **Method**              | **Harmlessness** | **Helpfulness** | **Summary** |
> |----------------|-------------------------|------------------|-----------------|-------------|
> | Gemma-1-2B     | CAI                     | 0.55             | 0.58            | 0.67        |
> | Gemma-1-2B     | RLCD                    | 0.61             | 0.67            | 0.72        |
> | Gemma-1-2B     | Conventional RLAIF      | 0.59             | 0.69            | 0.71        |
> | Gemma-1-2B     | Curriculum-RLAIF        | **0.68**             | **0.72**            | **0.79**        |
> | |
> | LLaMA-3-8B     | CAI                     | 0.57             | 0.62            | 0.70        |
> | LLaMA-3-8B     | RLCD                    | 0.65             | 0.77            | 0.78        |
> | LLaMA-3-8B     | Conventional RLAIF      | 0.71             | 0.76            | 0.82        |
> | LLaMA-3-8B     | Curriculum-RLAIF        | **0.77**             | **0.81**            | **0.89**        |
>
> These results indicate that Curriculum-RLAIF achieves higher preference annotation accuracy than competing methods when evaluated against *human-labeled ground truth*.
>
> For reference, a comprehensive empirical comparison of LLM-based against human-based preference annotation for alignment was conducted by Lee et al., “RLAIF vs. RLHF: Scaling Reinforcement Learning from Human Feedback with AI Feedback,” ICML 2025. The RLAIF variant they proposed, which is referred to as “Conventional RLAIF” in our paper, serves as a baseline in our manuscript.
>
> **W2:** performance comparison between Curriculum-RLAIF and baseline methods
>
> **R2:** Thank you for the insightful comments regarding the differences between Tables 1 and 2.
> The performance of the baseline curriculum methods in Table 1 differs from that in Table 2 because the evaluations use different data sources, as indicated in the second column of Table 2. The best-performing methods in Table 2 rely on ***data sources constructed by Curriculum-RLAIF***.
>
> We conducted this ablation study to investigate the impact of the data. The results show that simply adopting the Curriculum-RLAIF data source, i.e., $D_{ctr} + D_{brg}^{+/-} + D_{rnd}$, can substantially boost existing methods. Although the best-performing enhanced baseline achieves performance comparable to Curriculum-RLAIF, our method incurs much lower computation cost, as analyzed in Table 4 of Appendix D.
>
> This improvement underscores the general merit of the automatically constructed data source produced by the Curriculum-RLAIF pipeline for RL-based LLM alignment. In addition to the data-source ablation, we present an ablation of Curriculum-RLAIF’s curriculum strategies in Section 5.3. Together, these ablations clarify the technical choices behind the final method, demonstrating that both the data resources and the curriculum design are crucial to Curriculum-RLAIF’s performance.
>
> **W3:** performance improvement over the baselines
>
> **R3:** We assume this concern stems from confusion about W2. We hope our explanation of the experimental setup, specifically the differences between the results in Table 1 and Table 2, has addressed this concern. We would be grateful for any further questions or concerns you may have.

---

> ### Author Response · Authors · 2025-11-27
> **Response to Reviewer 59Mb (2/2)**
>
> **W4:** Typo in table 1 \
> **R4:** Thank you for your careful reading. We have removed the typo and performed additional proofreading for paper quality.
>
> **W5:** potential catastrophic forgetting from distinct non-overlapping chunks of the dataset in the Internal evaluation method, and the technical choice of dataset merging in Curriculum-RLAIF. \
> **R5:** The curriculum training does not merge the new dataset. The potential “catastrophic forgetting” issue is very interesting, and could serve as a possible hypothesis to the research question we discussed in Section 6. Below is a quote of the discussion for your convenience:
>
> > As we see in Fig.3 and Fig.5, our curriculum at each stage includes samples spanning a broader range of difficulty levels, yet achieves comparable or even superior performance compared to the internal evaluation baseline. This suggests that overly strict data selection based on data difficulty may not be an optimal curriculum design. Instead, incorporating samples with a moderate range of difficulty at each stage may serve as an effective regularization strategy to enhance generalizability [1, 2].
>
> We hypothesized that the internal evaluation should perform even better, however, it does not. Maybe as you pointed out, some degree of “forgetting” may limit it, while the data “overlapping” induced by our method appears to mitigate the effect. We believe this is a very interesting research question to study.
>
> Reference: \
> [1] Srivastava et al., Dropout: A Simple Way to Prevent Neural Networks from Overfitting, JMLR. 2014. \
> [2] Alex Hernández-García and Peter König, Data augmentation instead of explicit regularization, arxiv, 2020
>
> We thank to reviewer 59Mb again for thoughtful questions and constructive feedback, which have helped us improve the manuscript. We hope our responses address your concerns. We welcome any further comments and remain available to discuss as needed.

---

### Note · Authors · 2026-01-05

I have read and agree with the venue's withdrawal policy on behalf of myself and my co-authors.